# Trends and patterns of second-hand smoke exposure amongst the non-smokers in India-A secondary data analysis from the Global Adult Tobacco Survey (GATS) I & II

Madhur Verma[1], Soundappan Kathirvel[2], Milan Das[3], Ramnika Aggarwal[1], Sonu Goel ®[2]*

1 Department of Community & Family Medicine, All India Institute of Medical Sciences, Bathinda, Punjab, India, 2 Department of Community Medicine and School of Public Health, Post Graduate Institute of Medical Education and Research, Chandigarh, India, 3 International Institute for Population Sciences, Mumbai, India

* sonugoel007@yahoo.co.in

**Data Availability Statement:** The authors confirm that, for approved reasons, some access restrictions apply to the data underlying the

## Abstract

### Objectives

The primary objective of the present study was to compare the prevalence and patterns of second-hand smoke (SHS) exposure in the home, workplace, public places, and at all three places amongst the non-smoker respondents between the two rounds of Global Adult Tobacco Survey (GATS) in India. The secondary objectives were to assess the differences in various factors associated with SHS exposure among non-smokers.

### Study design

This secondary data analysis incorporated data generated from the previous two rounds of the cross-sectional, nationally representative GATS India, which covered 69,296 and 74,037 individuals aged 15 years and above. Exposure to the SHS at home, workplace, and public places amongst the non-smokers were the primary outcome variables. Standard definitions of the surveys were used.

### Results

The overall weighted prevalence of exposure to SHS amongst the non-smokers inside the home and public places reduced. In contrast, the prevalence in the workplace increased marginally in round II compared to I. The proportion of adults who were exposed to SHS at all three places did not change much in two rounds of surveys. A decrease in the knowledge of the respondents exposed to SHS at home and public places was observed about the harmful effects of smoking in round II. Age, gender, occupation, place, and region of respondents were found to be significant determinants of SHS exposure at all the three places on multinomial logistic regression analysis.

### Conclusions

The study calls for focused interventions in India and stringent implementation of anti-tobacco legislation, especially in the workplaces for reducing the exposure to SHS amongst

findings. Data are from the Global Adult Tobacco Survey India Study and are conducted by the International Institute for Population Sciences (www.iipsindia.org). Data can be obtained by filling out a data request form available at the institute website and paying appropriate charges. Since this data is obtained from the International Institute for Population Sciences for this paper by the authors, we cannot directly give the data to any other researcher. Please find the direct link for the data request form in International Institute for Population Sciences Website: http://www.iipsindia.org/pdf/Data_Request_Form.pdf.

**Funding:** The author(s) received no specific funding for this work.

**Competing interests:** The authors have declared that no competing interests exist.

the non-smokers and to produce encouraging and motivating results by next round of the survey.

## Introduction

Second-hand smoke (SHS) is expressed as the "sum of tobacco smoke exposures in the multiple microenvironments where a person spends time" [1]. The SHS concentration depends on the number of tobacco products smoked during a period- the volume of the enclosed space, the ventilation rate, other processes that might eliminate pollutants, and individual-related characteristics [2]. The exposure mainly consists of the smoke released from the burning end of a smoldering cigarette, pipe, or cigar ("side-stream smoke," 85%) and, to a lesser extent, the smoke exhaled from the lungs of an active smoker nearby ("mainstream smoke," 15%) [3]. Chronic exposure to SHS is suggested to be, on average, 80%–90% as harmful as chronic active smoking with a significant dose-response relationship [4–6]. SHS exposure is attributed to the same complications as active smoking, including both acute and chronic diseases [2]. For instance, SHS affects the heart and blood vessels, increasing the risk of myocardial infarction, stroke, and emotional changes like depression in non-smokers. In very young children, SHS increases the risk for more severe problems, including sudden infant death syndrome [7]. There is a high incidence of tuberculosis among smokers and those exposed to SHS [8].

As per the Global burden of disease estimates, the three leading risk factors for global disease burden were high blood pressure, tobacco smoking including second-hand smoke (SHS), and household air pollution from solid fuels [9]. Each year, more than 1,200,000 deaths amongst the non-smokers result from exposure to SHS around the world [10]. There is a high burden of SHS in different countries of the world. Home, workplace, and public places are designated as the most common places for SHS exposure [11]. As per the GATS Atlas report 2015, around 392 million adults are exposed to SHS in their workplace, while 1.5 billion people are exposed to SHS at home in the 22 Global Adult Tobacco Survey (GATS) countries [12]. In Bangladesh, China, and Egypt, 60% or more of adults who work indoors have been exposed to SHS at their workplace [12]. The exposure to SHS in the workplace ranges between 17.0% and 61.0%, while SHS at home is between 12.6% and 72.4% [13]. In addition, approximately 1.2 billion people >15 years of age are exposed to SHS in public places in 22 GATS countries. SHS in public places is highest in restaurants among the four public venues, i.e., government buildings, healthcare facilities, restaurants, & public transportation, and ranges between 4% in Uruguay and 88% in China.

Smoke-free laws are essential to check exposure to SHS. To protect the non- smokers from SHS- Article 8 of WHO Framework Convention on Tobacco Control (WHO FCTC) stipulates that "tobacco smoke be eliminated from all indoor workplaces- indoor public places- public transport- and as appropriate- in other public places" [14]. In concordance with global FCTC guidelines- India has also implemented its national legislation named 'The Cigarettes and Other Tobacco Products Act- 2003 (COTPA)'- Section 4 of which entails the prohibition of smoking in public places [15]. As a result- about 60%–70% of public places are smoke-free in India- as per the WHO Global tobacco epidemic 2017 report [16].

The risk of SHS exposure among non-smokers in India is relatively high. GATS I (2009–2010) of India reported that 48.0% and 26.1% of respondents were exposed to SHS in the home and workplace, respectively. Subsequently, the GATS II (2016–17) of India revealed that the proportion of non-smokers exposed to SHS has decreased significantly from **48% in**

**GATS I to 35.0%** at home and from 29% in GATS I to 25.7% in GATS II a public place [17,18]. In contrast, the proportion of the non-smokers working in closed indoor areas exposed to SHS has increased, though non-significant from **26.1% in GATS I to 26.2%** in GATS II [17,18].

Although there is an overall significant reduction in SHS exposure in-home and public places, limited information is available at the subgroup level and factors associated with this changing pattern of SHS exposure. Similarly, limited information is available on negligible change in SHS exposure in the workplace despite the ban in public and workplaces. With this background, we did secondary data analysis, with the primary objective of comparing the prevalence and patterns of SHS exposure in the home, workplace, and public places amongst the non-smoker respondents between the two rounds of GATS in India. The secondary objectives were to estimate the proportion of the people who are exposed to SHS at all three places and various factors associated with this kind of exposure.

## Methods

### Data source

This secondary data analysis incorporated data generated from the previous two rounds of the cross-sectional, nationally representative survey known as the GATS-1 (2009–10) and GATS-II (2016–17). The Ministry of Health and Family Welfare (MoHFW), Government of India, designated the International Institute for Population Sciences, as the nodal agency for conducting GATS-I survey & Tata Institute of Social Sciences (TISS), as the nodal implementing agency for the GATS-II survey. All Indian states were included in GATS-I and GATS-II. A standard protocol concerning the questionnaire, sample design, data collection, and management procedures was used in these surveys [17,18].

Both surveys covered domains like tobacco use (smoking and smokeless tobacco), exposure to second-hand smoke, cessation, the economics of tobacco, exposure to media messages on tobacco use and knowledge, attitude, and perceptions towards tobacco use. The GATS-I survey provided the baseline estimates of the prevalence of tobacco use and key indicators relevant to the National Tobacco Control Program. GATS-II was designed to measure the changes in key indicators of the tobacco control program.

### Operational definitions and sample selection

Standard operational definitions and protocols were used as per the GATS methodology to identify the sample for our analysis. Figs 1 and 2 depicts the process of the sample selection from GATS I and II.

**SHS exposure at home** was estimated for non-smokers who reported anyone smoking inside his/her home (that excludes areas outside,- such as patios, balcony, garden, etc. that are not fully enclosed). SHS was taken to be present ("Yes") if the response to the question "*How often does anyone smoke inside your home*?" was 'more than or equal to once in the last one month' [17–19].

**SHS exposure at the workplace** was assessed for the respondents who work outside of the home and who usually work indoors or both indoors and outdoors. SHS is defined as the percentage of respondents who reported someone smoking at least in indoor workplaces in the past 30 days before the survey. SHS was considered to be present ("Yes") if the response to the question "During the past 30 days- did anyone smoke in indoor areas where you work?" was affirmative.

**SHS exposure at public places** (government building, healthcare facility, private offices, restaurant, public transportation, night club, and cinema hall) is estimated for non-smokers

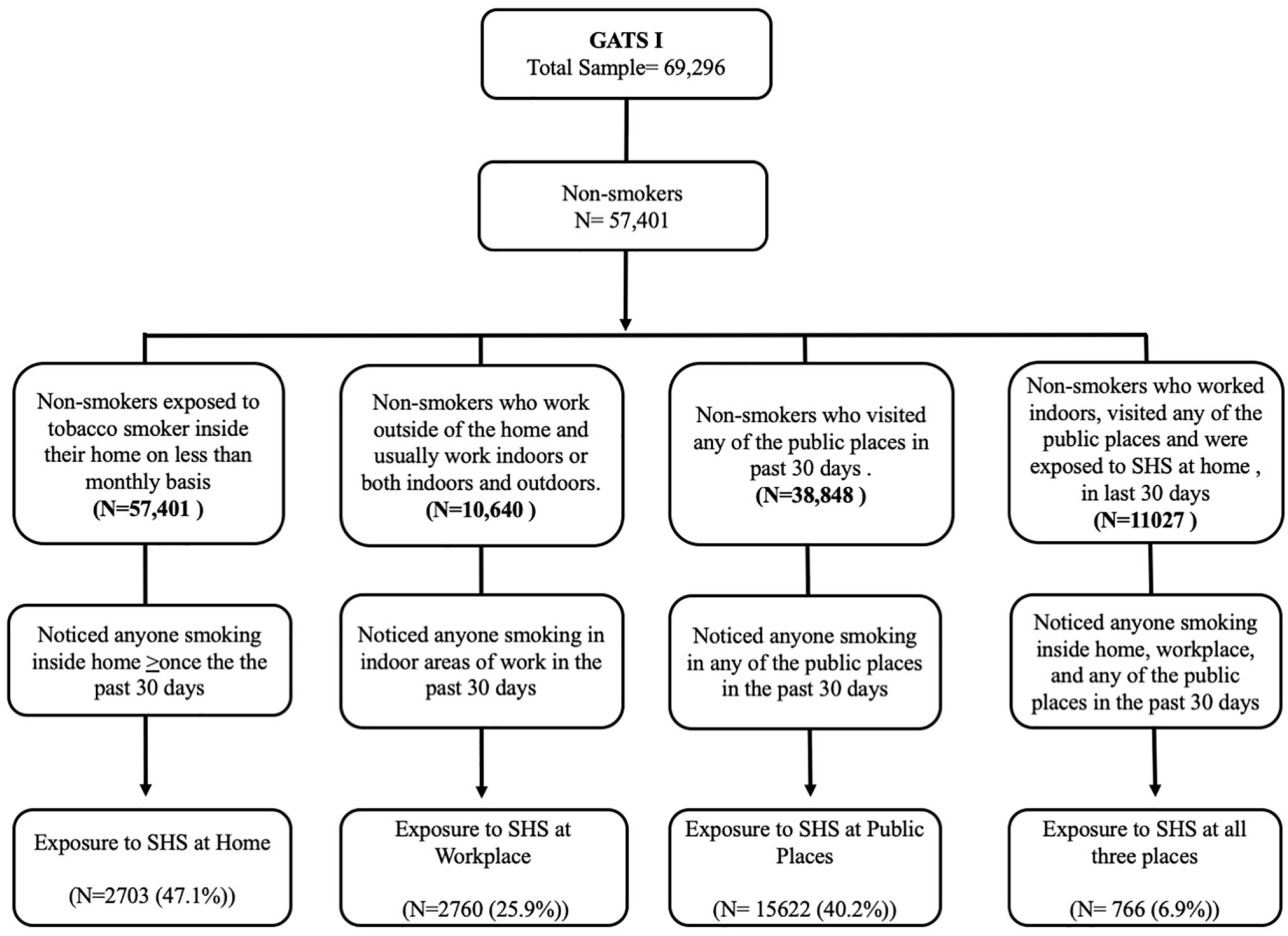

**Fig 1. Exposure to second-hand smoke among the non-smokers in 1st round of Global Adult Tobacco Survey (2009–10).**

respondents who reported someone smoking inside the public places of interest, in the past 30 days and responded exposed to smoke at any one of these places at least once.

**SHS exposure at all three places** (home, workplace, and any of the public places) was estimated by estimating the number of non-smokers who were exposed to SHS at the home, workplace, and any of public places in the last 30 days.

The household economic status was measured using the household assets information provided by the survey. Since household information of assets was relatively inadequate in nature, the scores of 10 household assets were summed up to give a final score between 0 and 10. These scores were disseminated into three parts based on their distribution, and households were categorized as poor, middle, and rich.

All States/Union Territories of India were divided into six geographical regions for analysis. The north region contains Jammu and Kashmir, Himachal Pradesh, Punjab, Chandigarh, Uttarakhand, Haryana, and Delhi; central includes Rajasthan, Uttar Pradesh, Chhattisgarh, and Madhya Pradesh; east contains West Bengal, Jharkhand, Odisha, and Bihar; northeast includes Sikkim Arunachal Pradesh, Nagaland, Manipur, Mizoram, Tripura, Meghalaya, and Assam; west contains Gujarat, Maharashtra, and Goa; south contains Andhra Pradesh (later divided into Andhra Pradesh and Telangana), Karnataka, Kerala, Tamil Nadu, and Puducherry.

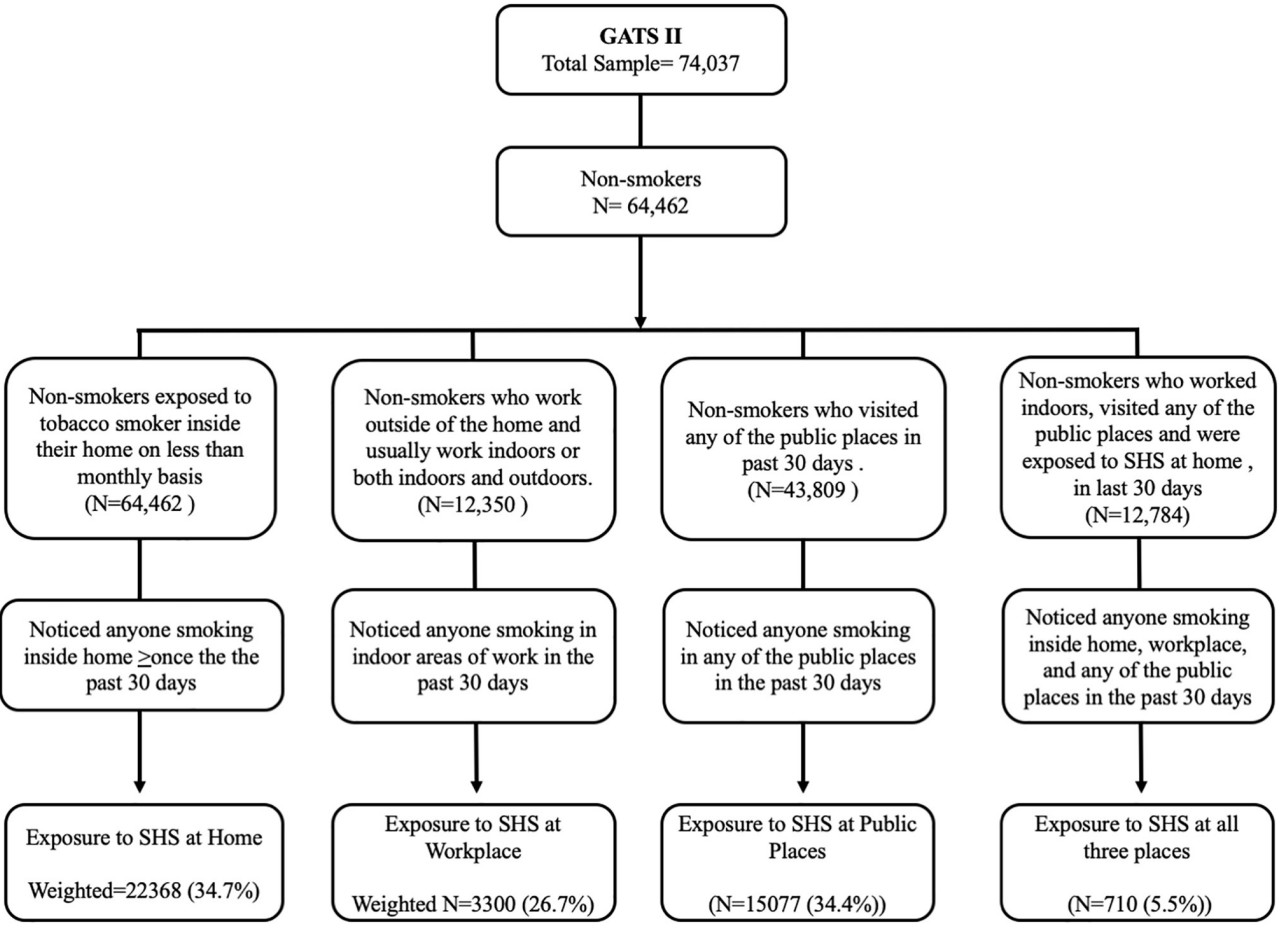

**Fig 2. Exposure to second-hand smoke among the non-smokers in 2nd round of Global Adult Tobacco Survey (2009–10).**

## Statistical analysis

Bivariate analysis was used to estimate the association between SHS exposure at home, work-place and public places, and socio-demographic characteristics. Further, multinomial logistic regression was used to estimate and assess the adjusted associations between SHS exposure at home, work-place, and public place as well as all the three places with various socio-demographic factors and factors pertaining to the knowledge regarding harmful effects of tobacco from GATS I and II. The socio-demographic factors used in the bivariate and multivariable logistic regression analysis are age (3 categories), sex (male/female), education (5 categories), occupation (5 categories), employment (4 categories), wealth (3 categories), place of residence (rural /urban) and region (6 categories). All knowledge related variables (that exposure to smoking cause stroke- heart attack- lung cancer—serious illness.) are used as a dichotomous form (yes/no). Each outcome was modeled relative to baseline outcome group: non-exposure to SHS at different places for reporting the adjusted prevalence ratios (PR). Models were used after applying the sampling weights and adjusting for multistage sampling designs using *svy* command in STATA version 13.0 that was used to carry out the statistical analysis. A p-value <0.05 was considered statistically significant.

## Results

A total of 69,296 and 74,037 non-smokers aged ≥15 years were included in GATS-1 and GATS-II, respectively. Of these, 57,401 (82.8%) & 64,462 (87.1%) non-smokers were included in the analysis for estimation of SHS prevalence at home, while 10,640 (15.4%) & 12,350 (16.7%) non-smokers who were working indoors were included in the analysis for estimation of SHS prevalence in the workplace in GATS-I and GATS-II respectively. Similarly, 38,848 (56.1%) & 43,809 (59.2%) non-smokers who visited any of the public places in the last 30 days were included for the estimation of SHS exposure at any of the public places and 10,640 (15.4%) & 12,350 (16.7%) non-smokers respondents who were included to estimate SHS exposure prevalence at all the three places in last 30 days in GATS-I and GATS-II respectively.

### Prevalence of exposure to second-hand smoking

Table 1 depicts the weighted prevalence with a 95% confidence interval of exposure to SHS at home, workplace, any of the public places, and all the three places. As per GATS-II, the highest prevalence of exposure to SHS was seen at home, followed by workplace and public places. Being female, illiterate, poor, homemaker, resident of the rural areas, and from the central region of India, associated with the highest exposure to SHS at home in GATS II. The overall prevalence of exposure to SHS amongst the non-smokers in-home reduced from 47.1% in GATS I to 34.7% in GATS II. This decrease in the prevalence of SHS exposure at home was observed across all age groups. (Table 1). There was a significant (p<0.05) decrease in the knowledge of respondents about the harmful effects of smoking from GATS-I to GATS-II.

Nearly one-fourth of all the respondents who worked outside their home and in indoor areas were exposed to SHS in the last 30 days in GATS-II. The prevalence of SHS exposure (Table 1) at the workplace amongst the non-smokers was significantly high in the younger age groups, male workers, less educated, and poorest respondents from rural areas of central India. The overall prevalence of SHS exposure at the workplace increased marginally in GATS II (26.7%) as compared to GATS I (25.9%). Workers also depicted an improvement in knowledge regarding the harmful effects of smoking in GATS II as compared to the GATS I Survey.

In GATS-II, nearly one-third of non-smokers were exposed to SHS at any of the public places in the last 30 days. The highest prevalence of exposure to SHS at any one of the public places was observed in males, younger age group, educated respondents, urban-areas residents, govt and non-govt employees, richest quintiles, and residents of central India. Prevalence in GATS II (40.2%) decreased, as compared to GATS I (40.2%). These respondents also depicted a decrease in knowledge regarding the harmful effects of smoking. The proportion of adults who were exposed to SHS at all three places in the last 30 days decreased from 6.9% in GATS I to 5.5% in GATS II.

### Factors affecting exposure to second-hand smoking

Table 2 depicts the adjusted prevalence ratio for SHS exposure at home and its association with socio-demographic and economic characteristics and knowledge on illness caused due to tobacco use. Younger age-group, female gender, illiteracy, residence in rural areas, most impoverished socioeconomic status had higher chances of exposure to SHS at home. Respondents who lacked knowledge regarding the harmful effects of smoking were more likely to exposed to SHS at home.

A similar analysis was done for highlighting factors affecting SHS exposure at the workplace. Youngest age group, i.e., 15–30 years, male gender, less educated, govt/Pvt employees, the residents from rural areas, most impoverished people, were more likely to be exposed to the SHS at the workplace at per GATS II (Table 3). In GATS I, North-eastern India had the

Table 1. Weighted prevalence of exposure to second-hand smoke at home, workplace, public places, and all three places among non-smokers as per GATS India round I&II.

| | Home | | | Workplace | | | Public places | | | All the places | | |
|---|---|---|---|---|---|---|---|---|---|---|---|---|
| | GATS I Prevalence in % (95% CI) | GATS II Prevalence in % (95%CI) | χ2 (p-Value) | GATS I Prevalence (95% CI) | GATS II Prevalence (95%CI) | χ2 (p-Value) | GATS I Prevalence (95% CI) | GATS II Prevalence (95%CI) | χ2 (p-Value) | GATS I Prevalence (95% CI) | GATS-II Prevalence (95%CI) | χ2 (p-Value) |
| Total (n) | 57,401 | 64,462 | | 10,640 | 12,350 | | 38,848 | 43,809 | | 10,640 | 12,350 | |
| The overall prevalence of SHS | 47.1% | 34.7% | | 25.9% | 26.7% | | 40.2% | 34.4% | | 6.9% | 5.5% | |
| **Age (in years) Groups** | | | | | | | | | | | | |
| 15–30 | 49.7 (49.2–50.3) | 37.1 (36.5–37.6) | 626.6 (<0.001) | 27.1 (25.9–28.3) | 29.1 (27.8–30.5) | 17.8 (<0.001) | 43.4 (42.7–44.1) | 37.3 (36.6–37.9) | 331.1 (<0.001) | 8.3 (7.5–9.0) | 6.8 (6.1–7.4) | 33.1 (<0.001) |
| 31–44 | 45.5 (44.8–46.3) | 34.1 (33.4–34.8) | | 25.5 (24.1–26.8) | 26.0 (24.7–27.3) | | 39.4 (38.5–40.3) | 34.4 (33.5–35.2) | | 6.2 (5.4–6.9) | 5.2 (4.6–5.9) | |
| >45 | 43.2 (42.3–44.0) | 31.8 (31.1–32.4) | | 23.8 (21.8–25.7) | 24.4 (22.9–25.9) | | 33.5 (32.4–34.5) | 29.9 (29.1–30.7) | | 4.9 (3.9–5.9) | 4.1 (3.4–4.7) | |
| **Gender** | | | | | | | | | | | | |
| Male | 43.9 (42.5–43.7) | 30.7 (30.2–31.3) | 820.8 (<0.001) | 27.9 (26.9–28.9) | 28.9 (27.9–29.8) | 124.7 (<0.001) | 48.8 (48.1–49.5) | 42.8 (42.1–43.4) | 2000 (<0.001) | 7.9 (7.3–8.4) | 6.2 (5.8–6.7) | 7.6 (<0.01) |
| Female | 50.4 (49.9–50.9) | 38.1 (37.6–38.6) | | 17.9 (16.3–19.6) | 17.8 (16.4–19.4) | | 30.5 (29.8–31.1) | 24.8 (24.2–25.4) | | 3.6 (2.9–4.4) | 2.8 (2.1–3.4) | |
| **Education** | | | | | | | | | | | | |
| No education | 57.5 (56.8–58.3) | 44.7 (43.9–54.5) | 3800 (<0.001) | 35.9 (32.9–38.8) | 33.7 (31.5–35.9) | 402.7 (<0.001) | 34.6 (33.6–35.6) | 28.5 (27.6–29.5) | 201.9 (<0.001) | 8.2 (6.7–9.8) | 7.9 (6.6–9.2) | 141.3 (<0.001) |
| Less than primary | 50.0 (49.8–50.5) | 37.5 (36.2–38.8) | | 33.9 (30.8–37.2) | 35.7 (32.5–38.8) | | 39.4 (37.9–40.9) | 31.1 (29.6–32.7) | | 8.5 (6.7–10.4) | 10.4 (8.4–12.4) | |
| primary school completed | 49.1 (48.3–49.9) | 37.5 (36.8–38.3) | | 30.4 (28.7–32.0) | 32.2 (30.6–33.8) | | 41.7 (40.8–42.6) | 34.5 (33.7–35.4) | | 9.4 (8.3–10.4) | 7.2 (6.3–8.1) | |
| Secondary and above | 33.5 (32.7–34.2) | 25.3 (24.7–25.9.) | | 20.7 (19.7–21.8) | 20.6 (19.6–21.6) | | 42.7 (41.9–43.6) | 37.6 (36.9–38.3) | | 5.2 (4.6–5.8) | 3.5 (3.1–3.9) | |
| **Employment** | | | | | | | | | | | | |
| Govt/Non-Govt employees | 43.1 (43.1–44.8) | 23.9 (22.9–24.9) | 630.7 (<0.001) | 23.4 (22.3–24.5) | 19.5 (18.4–20.5) | 454.3 (<0.001) | 44.4 (43.4–45.4) | 41.3 (40.1–42.6) | 1100 (<0.001) | 7.0 (6.4–7.7) | 4.3 (3.7–4.8) | 65.2 (<0.001) |
| Self-employed | 48.8 (48.9–50.5) | 36.3 (37.6–38.9) | | 30.8 (29.4–32.2) | 33.6 (31.9–35.2) | | 43.8 (42.8–44.8) | 37.6 (36.8–38.3) | | 7.4 (6.6–8.3) | 6.7 (6.1–7.3) | |
| Student | 43.8 (42.7–44.9) | 31.3 (30.3–32.3) | | 13.8 (11.3–16.3) | 31.3 (29.7–32.9) | | 44.9 (43.6–46.2) | 39.6 (38.0–40.4) | | 2.7 (1.5–3.9) | 4.4 (2.9–5.8) | |
| Homemakers | 50.0 (49.5–50.9) | 38.3 (37.6–38.9) | | Not applicable | Not applicable | | 30.9 (30.0–31.8) | 24.3 (23.5–25.0) | | 7.8 (4.4–11.3) | 2.7 (0.8–4.5) | |
| Unemployed | 45.2 (42.8–46.4) | 32.6 (31.1–34.1) | | 47.1 (40.4–53.7) | 27.6 (23.9–31.3) | | 37.2 (34.9–39.5) | 31.1 (29.3–33.0) | | 9.1 (5.3–12.9) | 9.7 (0.6–13.4) | |
| **Place of residence** | | | | | | | | | | | | |
| Urban | 33.8 (37.0–38.5) | 24.7 (24.2–25.3) | 3000 (<0.001) | 24.3 (23.1–25.4) | 22.6 (22.2–23.7) | 161.7 (<0.001) | 38.9 (38.1–39.7) | 34.5 (33.8–35.2) | 35.3 (<0.001) | 5.5 (4.9–6.1) | 4.3 (3.8–4.8) | 92.9 (<0.001) |
| Rural | 52.9 (52.4–53.4) | 40.2 (39.6–40.6) | | 27.6 (26.4–28.8) | 30.5 (29.4–31.7) | | 40.9 (40.3–41.5) | 34.3 (33.8–34.9) | | 8.4 (7.7–9.1) | 6.7 (6.1–7.3) | |
| **Wealth index** | | | | | | | | | | | | |

(Continued)

**Table 1.** (Continued)

| | Home | | | Workplace | | | Public places | | | All the places | | |
|---|---|---|---|---|---|---|---|---|---|---|---|---|
| | GATS I Prevalence in % (95% CI) | GATS II Prevalence in % (95%CI) | χ2 (p-Value) | GATS I Prevalence (95% CI) | GATS-II Prevalence (95%CI) | χ2 (p-Value) | GATS I Prevalence (95% CI) | GATS-II Prevalence (95%CI) | χ2 (p-Value) | GATS I Prevalence (95% CI) | GATS-II Prevalence (95%CI) | χ2 (p-Value) |
| Poor | 54.6 (53.9–55.3) | 40.4 (39.7–41.0) | 1800 (<0.001) | 30.7 (29.2–32.2) | 29.4 (28.0–30.7) | 173.7 (<0.001) | 40.5 (39.7–41.3) | 34.6 (33.9–35.4) | 13.1 (<0.001) | 8.4 (7.5–9.2) | 6.3 (5.6–7.0) | 36.9 (<0.001) |
| Middle | 47.7 (47.8–48.2) | 34.7 (34.0–35.3) | | 24.7 (23.2–26.1) | 27.7 (26.3–29.0) | | 40.4 (39.5–41.3) | 33.1 (32.3–33.9) | | 6.7 (5.9–7.5) | 4.9 (4.2–5.5) | |
| Rich | 38.2 (37.5–38.9) | 28.5 (27.9–29.1) | | 22.2 (20.9–23.6) | 22.7 (21.4–24.0) | | 39.7 (38.9–40.6) | 35.5 (34.7–36.3) | | 5.7 (4.9–6.4) | 5.4 (4.7–6.1) | |
| **Region** | | | | | | | | | | | | |
| North | 44.6 (42.7–46.3) | 42.6 (41.2–43.9) | 9300 (<0.001) | 19.7 (17.0–22.3) | 26.8 (24.3–29.2) | 164.9 (<0.001) | 37.4 (35.3–39.5) | 37.0 (35.5–38.6) | 1200 (<0.001) | 6.8 (5.2–8.4) | 8.0 (6.6–9.4) | 393.2 (<0.001) |
| Central | **62.3 (61.5–62.9)** | **50.9 (50.3–51.7)** | | 25.7 (23.9–27.4) | 31.9 (30.3–33.5) | | 51.7 (50.8–52.7) | 44.3 (43.4–45.2) | | 12.6 (11.5–14.2) | 10.9 (9.9–11.9) | |
| East | 55.2 (54.4–56.2) | 38.2 (37.4–38.9) | | 29.9 (27.6–32.1) | 28.1 (26.1–29.9) | | 37.9 (20.91–22.4) | 32.7 (31.7–33.7) | | 6.4 (5.3–7.6) | 5.7 (4.7–6.7) | |
| North East | 48.6 (46.3–50.8) | 42.6 (40.5–44.6) | | 32.2 (27.5–36.9) | 26.2 (22.2–30.1) | | 36.0 (33.4–38.6) | 25.90 (23.5–28.2) | | 8.4 (5.8–10.9) | 4.4 (2.5–6.2) | |
| West | 39.3 (38.1–40.0) | 23.4 (22.6–24.2) | | 24.1 (22.3–25.8) | 19.6 (17.9–21.3) | | 40.1 (38.9–41.2) | 32.7 (31.6–33.8) | | 7.2 (6.1–8.2) | 1.9 (1.3–2.5) | |
| South | 24.6 (23.9–25.3) | 13.9 (13.3–14.4) | | 26.4 (24.9–27.9) | 24.8 (23.4–26.4) | | 31.7 (30.8–32.6) | 26.3 (25.4–27.1) | | 2.5 (1.9–3.1) | 0.9 (0.6–1.2) | |
| **The knowledge that smoking causes** | | | | | | | | | | | | |
| Stroke | 45.2 (44.6–45.7) | 32.6 (32.2–33.1) | 372.2 (<0.001) | 24.0 (22.9–25.1) | 27.1 (26.2–28.0) | 13.5 (<0.001) | 43.2 (42.6–43.9) | 35.7 (35.1–36.2) | 13.4 (<0.001) | 6.8 (6.2–7.4) | 4.9 (4.5–5.4) | 11.2 (<0.001) |
| Heart attack | 44.8 (44.3–45.3) | 32.7 (32.3–33.2) | 513.4 (<0.001) | 23.9 (23.0–24.9) | 26.4 (25.5–27.2) | 44.9 (<0.001) | 41.8 (41.2–42.3) | 35.4 (34.9–35.9) | 2.5 (>0.05) | 6.5 (5.9–7.1) | 5.3 (4.9–5.7) | 10.1 (<0.01) |
| Lung cancer | 46.1 (45.6–46.5) | 34.5 (34.1–34.9) | 165.2 (<0.001) | 25.3 (24.5–26.2) | 26.8 (25.9–27.6) | 4.3 (<0.5) | 41.4 (40.5–41.5) | 34.9 (34.9–35.9) | 47.9 (<0.001) | 6.9 (6.4–7.4) | 5.6 (5.2–5.9) | 0.2 (>0.05) |
| Serious illness | 46.5 (46.0–46.9) | 34.6 (34.2–34.9) | 6.11 (<0.01) | 25.9 (25.1–26.8) | 26.7 (26.4–28.0) | 0.9 (>0.05) | 40.6 (40.5–41.5) | 34.5 (34.4–35.4) | 11.9 (<0.001) | 6.8 (6.3–7.3) | 5.6 (5.2–6.0) | 3.2 (>0.05) |

**Table 2. Factors affecting second-hand smoke exposure among the non-smokers at home as per GATS India round I& II.**

| | SHS exposure at home Adjusted Prevalence Ratio (95% CI) | |
|---|---|---|
| | GATS-I | GATS–II |
| **Age Groups** | | |
| Between 15–30 years | 1.4 (1.3–1.4) | 1.5 (1.4–1.5) |
| Between 31–45 years | 1.2 (1.1–1.2) | 1.2 (1.1–1.2) |
| Above 45 years | 1 | 1 |
| **Sex** | | |
| Male | 1 | 1 |
| Female | 1.3 (1.2–1.3) | 1.5 (1.5–1.6) |
| **Education** | | |
| No formal education | 1.7 (1.6–1.8) | 1.8 (1.7–1.9) |
| Less than primary | 1.8 (1.7–1.9) | 1.7 (1.6–1.8) |
| Primary but less than secondary | 1.6 (1.6–1.7) | 1.5 (1.5–1.6) |
| Secondary and above | 1 | 1 |
| **Employment** | | |
| Govt/ Pvt. Employees | 1 | 1 |
| Self employed | 1.1 (1.1–1.2) | 1.3 (1.2–1.4) |
| Student | 1.0 (0.9–1.1) | 1.1 (0.9–1.2) |
| Homemakers | 1.0 (0.9–1.0) | 1.0 (0.9–1.1) |
| Unemployed | 1.1 (1.0–1.2) | 1.1 (1.0–1.2) |
| **Place of residence** | | |
| Urban | 1 | 1 |
| Rural | 1.7 (1.7–1.8) | 1.6 (1.5–1.7) |
| **Wealth-index quintile** | | |
| Poor | 1.6 (1.5–1.7) | 1.4 (1.4–1.5) |
| Middle | 1.4 (1.4–1.5) | 1.3 (1.2–1.3) |
| Rich | 1 | 1 |
| **Region** | | |
| North | 5.6 (5.2–5.9) | 5.9 (5.6–6.4) |
| Central | 5.0 (4.7–5.4) | 5.6 (5.2–5.9) |
| East | 3.6 (3.4–3.9) | 3.3 (3.1–3.5) |
| North East | 3.6 (3.4–3.9) | 6.7 (6.3–7.2) |
| West | 2.1 (2.0–2.3) | 2.0 (1.9–2.2) |
| South | 1 | 1 |
| **Knowledge smoking cause stroke** | | |
| Yes | 1 | 1 |
| No | 1.1 (1.1–1.2) | 1.1 (1.0–1.1) |
| **Knowledge smoking cause a heart attack** | | |
| Yes | 1 | 1 |
| No | 0.9 (0.9–1.0) | 1.1 (0.9–1.1) |
| **Knowledge of smoking lung cancer** | | |
| Yes | 1 | 1 |
| No | 1.0 (0.9–1.1) | 0.9 (0.9–1.0) |
| **Knowledge smoking causes serious illness** | | |
| Yes | 1 | 1 |
| No | **1.2 (1.1–1.2)** | **1.2 (1.1–1.2)** |

**Table 3. Factors affecting second-hand smoke exposure among non-smokers at the workplace as per GATS India rounds I& II, India.**

| | SHS exposure at Workplace Adjusted Prevalence Ratio (95% CI) | |
|---|---|---|
| | **GATS-I** | **GATS–II** |
| **Age Groups** | | |
| Between 15–30 years | 1.2 (1.1–1.4) | 1.2 (1.0–1.3) |
| Between 31–45 years | 1.1 (0.9–1.2) | 1.1 (0.9–1.2) |
| Above 45 years (**Ref**) | 1 | 1 |
| **Sex** | | |
| Male | 1.5 (1.3–1.6) | 1.8 (1.6–1.9) |
| Female | 1 | 1 |
| **Education** | | |
| No formal education | 1.6 (1.4–1.9) | 1.6 (1.4–1.9) |
| Less than primary | 1.7 (1.4–2.0) | 1.6 (1.5–2.0) |
| Primary but less than secondary | 1.4 (1.2–1.6) | 1.5 (1.3–1.6) |
| Secondary and above | 1 | 1 |
| **Employment** | | |
| Govt/ Pvt. Employees | 1.5 (1.3–1.6) | 1.7 (1.5–1.9) |
| Self employed | 0.8 (0.6–1.1) | 1.7 (1.5–1.9) |
| Student | 2.1 (1.5–2.8) | 1.4 (1.1–1.8) |
| Others | 1 | 1 |
| **Place of residence** | | |
| Urban | 1 | 1 |
| Rural | 1.1 (0.9–1.2) | 1.3 (1.1–1.4) |
| **Wealth-index quintile** | | |
| Poor | 1.2 (1.1–1.4) | 1.1 (1.0–1.3) |
| Middle | 1.1 (0.9–1.2) | 1.1 (1.0–1.2) |
| Rich | 1 | 1 |
| **Region** | | |
| North | 0.8 (0.7–0.9) | 1.2 (1.0–1.3) |
| Central | 1.0 (0.9–1.2) | 1.1 (0.9–1.2) |
| East | 0.9 (0.8–1.1) | 0.9 (0.8–1.1) |
| North East | 1.7 (1.4–1.9) | 1.1 (0.9–1.2) |
| West | 0.7 (0.6–0.9) | 0.8 (0.6–0.9) |
| South | 1 | 1 |
| **Knowledge smoking cause stroke** | | |
| Yes | 1 | 1 |
| No | 1.0 (0.9–1.1) | 0.9 (0.8–1.0) |
| **Knowledge smoking cause a heart attack** | | |
| Yes | 1 | 1 |
| No | 1.3 (1.1–1.4) | 1.0 (0.9–1.2) |
| **Knowledge of smoking lung cancer** | | |
| Yes | 1 | 1 |
| No | 0.8 (0.6–1.1) | 1.3 (0.6–1.0) |
| **Knowledge smoking causes serious illness** | | |
| Yes | 1 | 1 |
| No | 1.2 (0.9–1.5) | 0.9 (0.7–1.0) |

highest exposure to SHS at the workplace, while in GATS II, north India was dominating the scene. Respondents who don't' know that smoking causes lung cancer had significantly more chances of being exposed to SHS at the workplace as per GATS II. Similarly, younger age, male gender, residence in central India depicted the highest likelihood while education status, employment status, place of residence, and knowledge status depicted no effect on exposure to SHS at any of the public places (Table 4). Further, only youngest respondents (between 15–30 years of age), males, fewer years of education, unemployed and self-employed, and residence in the rural areas, and the northern part of India depicted highest chances of being exposed to SHS at all the three places, while wealth and knowledge regarding the harmful effects of smoking were not seen as significant factors as per the second rounds of GATS (Table 5).

## Discussion

Our analysis highlights the changes in the exposure to SHS among non-smokers in the home, workplace, and public places between two (nation-wide) surveys. Despite the fact that there was minimal change in the prevalence of exposure to SHS in the workplace in GATS II as compared to GATS I, an encouraging reduction in SHS exposure in-home and public places was observed. The disaggregated individual-level data analysis from nationally representative datasets along with an estimation of the prevalence of SHS exposure among non-smokers at all the three places is the main strength of this manuscript. In addition, we have assessed the factors associated with the change in prevalence of exposure to SHS at home, workplace, and public places over a period.

The overall prevalence of SHS exposure at home among the non-smokers in India was estimated at 34.7% as per GATS II, while the overall prevalence of SHS exposure at the workplace and public places was estimated as 26.7% and 34.4% respectively. The prevalence of SHS exposure in the home, workplace and any of the public places in 22 countries that participated in latest rounds of GATS ranged between 3.7% in Panama to 78.2% in Indonesia at home, 5.7% in Panama to 63.4% in China at the workplace, and 4% in Qatar, Mexico and Uruguay to 88% in China at any of the public places [12].

SHS exposure has significantly decreased amongst the non-smokers in the home from GATS I to II. However, SHS exposure is highest at home compared to other places of exposure anytime. This is especially making the vulnerable population, i.e., women, children, and elderly non-smokers, at high risk of exposure and hence to diseases. A review article by Mckay et al. also suggested that exposure at home (40%) was more common than exposure at work (29.9%) [20]. India was amongst the first signatory to the World Health Organization (WHO) Framework Convention on Tobacco Control (FCTC). The country has implemented The Cigarettes and Other Tobacco Products Act (COTPA), 2003, towards reducing tobacco use and protecting people from the dangers of SHS, especially banning at the workplace and public places. Though the COTPA 2003 doesn't ban smoking at home, the reduction in SHS at home over a period can be attributed to a phenomenon described as "norm spreading" and reduced social acceptability. [21,22] Apart from banning smoking in workplaces, a plethora of initiatives have been taken by the Government of India that focussed on the challenges highlighted by GATS-I in 2009–10. Under the National Tobacco Control Program (NTCP), substantial investments were made on a national level public awareness campaign, which helped in reaching out to varied audiences. In addition, a toll-free National Tobacco *Quitline* has been established under the National Tobacco Control Programme and m-Cessation services under the "Be Healthy Be Mobile Initiative." Further, the emphasis was laid on integrating tobacco -cessation in the healthcare delivery system by encouraging healthcare institutes to set up tobacco cessation facilities.

**Table 4. Factors affecting second-hand smoke exposure among the non-smokers at any of the public places as per GATS I& II, India.**

| | SHS exposure at any Public place Adjusted Prevalence Ratio (95% CI) | |
| --- | --- | --- |
| | GATS-I | GATS–II |
| **Age Groups** | | |
| Between 15–30 years | 1.3 (1.2–1.4) | 1.2 (1.2–1.3) |
| Between 31–45 years | 1.2 (1.1–1.2) | 1.1 (1.1–1.2) |
| Above 45 years | 1 | 1 |
| **Sex** | | |
| Male | 1.7 (1.6–1.8) | 1.8 (1.7–1.9) |
| Female | 1 | 1 |
| **Education** | | |
| No formal education | 0.9 (0.8–1.0) | 0.9 (0.9–1.0) |
| Less than primary | 0.9 (0.8–1.0) | 1.0 (0.9–1.1) |
| Primary but less than secondary | 0.9 (0.8–1.0) | 1.0 (0.9–1.0) |
| Secondary and above | 1 | 1 |
| **Employment** | | |
| Govt/ Pvt. Employees | 1 | 1 |
| self employed | 1.0 (0.9–1.1) | 1.0 (0.9–1.1) |
| Student | 1.1 (0.9–1.1) | 0.9 (0.8–1.0) |
| Homemaker | 0.9 (0.8–1.0) | 0.8 (0.8–0.9) |
| Unemployed | 0.8 (0.7–0.9) | 0.8 (0.7–0.9) |
| **Place of residence** | | |
| Urban | 1 | 1 |
| Rural | 1.2 (1.1–1.2) | 1.0 (0.9–1.1) |
| **Wealth-index quintile** | | |
| Poor | 0.9 (0.9–1.0) | 1.1 (0.9–1.0) |
| Middle | 1.0 (0.9–1.1) | 0.9 (0.9–1.0) |
| Rich | 1 | 1 |
| **Region** | | |
| North | 1.4 (1.3–1.5) | 1.6 (1.5–1.7) |
| Central | 2.5 (2.3–2.7) | 2.1 (2.0–2.3) |
| East | 1.5 (1.4–1.6) | 1.4 (1.3–1.6) |
| North East | 1.7 (1.5–1.8) | 1.3 (1.2–1.4) |
| West | 1.3 (1.2–1.4) | 1.3 (1.2–1.4) |
| South | 1 | 1 |
| **Knowledge smoking cause stroke** | | |
| Yes | 1 | 1 |
| No | 1.0 (0.9–1.1) | 0.9 (0.8–1.0) |
| **Knowledge smoking cause a heart attack** | | |
| Yes | 1 | 1 |
| No | 1.1 (0.1–1.1) | 0.9 (0.9–1.0) |
| **Knowledge of smoking lung cancer** | | |
| Yes | 1 | 1 |
| No | 0.9 (0.8–1.1) | 0.8 (0.7–0.9) |
| **Knowledge smoking causes serious illness** | | |
| Yes | 1 | 1 |
| No | 0.9 (0.8–1.1) | 1.1 (0.9–1.2) |

**Table 5. Factors affecting second-hand smoke exposure among the non-smokers at all three places (home, work-place, and any of the public places), as per GATS I& II, India.**

| | SHS exposure at all three places Adjusted Prevalence Ratio (95% CI) | |
| --- | --- | --- |
| | GATS-I | GATS–II |
| **Age Groups** | | |
| Between 15–30 years | 1.3 (1.0–1.6) | 1.4 (1.1–1.7) |
| Between 31–45 years | 1.1 (0.8–1.3) | 1.2 (0.9–1.4) |
| Above 45 years | 1 | 1 |
| **Sex** | | |
| Male | 1.0 (0.9–1.2) | 1.5 (1.2–1.8) |
| Female | 1 | 1 |
| **Education** | | |
| No formal education | 1.4 (1.1–1.9) | 1.6 (1.2–2.1) |
| Less than primary | 1.6 (1.2–2.2) | 2.5 (1.9–3.4) |
| Primary but less than secondary | 1.6 (1.3–1.9) | 1.7 (1.4–2.1) |
| Secondary and above | 1 | 1 |
| **Employment** | | |
| Govt/ Pvt. Employees | 1 | 1 |
| self employed | 1.2 (1.0–1.4) | 1.5 (1.3–1.9) |
| Student | 0.7 (0.4–1.0) | 1.1 (0.7–1.8) |
| Homemaker | 0.7 (0.5–1.1) | 1.3 (0.8–2.2) |
| Unemployed | 1.1 (0.7–1.8) | 1.9 (1.1–3.1) |
| **Place of residence** | | |
| Urban | 1 | 1 |
| Rural | 1.6 (1.2–1.7) | 1.3 (1.1–1.5) |
| **Wealth-index quintile** | | |
| Poor | 1.4 (1.2–1.8) | 0.8 (0.6–1.0) |
| Middle | 1.2 (0.9–1.5) | 0.9 (0.7–1.1) |
| Rich | 1 | 1 |
| **Region** | | |
| North | 5.6 (3.9–8.1) | 10.6 (6.7–16.9) |
| Central | 7.9 (5.4–11.7) | 12.4 (7.7–19.9) |
| East | 3.2 (2.1–5.0) | 6.4 (3.8–10.6) |
| North East | 7.2 (4.9,–10.4) | 9.1 (5.6–14.5) |
| West | 3.3 (2.2–4.9) | 3.4 (1.9–5.9) |
| South | 1 | 1 |
| **Knowledge smoking cause stroke** | | |
| Yes | 1 | 1 |
| No | 1.1 (0.9–1.4) | 1.1 (0.9–1.4) |
| **Knowledge smoking cause a heart attack** | | |
| Yes | 1 | 1 |
| No | 0.9 (0.8–1.2) | 0.9 (0.7–1.2) |
| **Knowledge of smoking lung cancer** | | |
| Yes | 1 | 1 |
| No | 1.1 (0.7–1.6) | 0.5 (0.3–0.9) |
| **Knowledge smoking causes serious illness** | | |
| Yes | 1 | 1 |
| No | 0.9 (0.5–1.4) | 0.5 (0.3–0.8) |

We observed that females were more likely to be exposed to SHS at home compared to males, and the chances are increased from GATS I to GATS II. This observation is in line with previous studies that depict similar concerns and depict more SHS exposure at home among females. [23–27] Singh and Lal stated that SHS is a gender issue as women are at a higher risk of SHS exposure [28]. In India, this risk can be attributed to the socio-cultural norms, where men are the bread-earners in most of the families, and women are less likely to protest against the husband's smoking behavior at home, intending to preserve familial congruity. [29–31] Since the women have no replacement substitute for being at home and cannot avoid exposure to SHS, it is a matter of concern. In addition, we observed that the younger non-smoker respondents, those from the rural areas, and socioeconomically disadvantaged households in terms of education and wealth index were more likely to be exposed to the SHS at home. These determinants are highly relevant, particularly in a country like India, where smoking is barely perceived as a risk factor for health as depicted by a low level of knowledge about the harmful effects of smoking, as elaborated in the later parts of the discussion. We should encourage the public to enact smoking policies that promote 'smoke-free homes.' Recent progress in developed countries has highlighted the feasibility of achieving smoke-free environments. It is high time that we learn from such experiences, taken to curb smoking in the home. [32] The interventions will help to reduce the initiation of tobacco use among non-smokers, especially the younger population.

Though we did not observe significant changes in SHS exposure at workplaces from GATS I to GATS II, the exposure is still high, i.e., one in four non-smokers still exposed at the workplace. This is rather demotivating, looking at the various public health interventions implemented by various state governments in addition to the implementation of COTPA 2003. As these efforts lack uniformity and multipronged strategies to control the demand [33], there is an urgent need for strict enforcement of anti-smoking laws in the workplace. Some workplaces, like hotels, take pride in just providing a ventilated smoking room as a part of their responsibility towards non-smokers. However, even the most advanced ventilation system cannot eliminate tobacco smoke or its risks. The employers should be enlightened that only 100% smoke-free environments are effective in protecting health, in addition to other benefits like lower health-care, cleaning and maintenance, and reduced fire, accident, and injury-related costs. Global data overwhelmingly confirm that going smoke-free does not harm businesses. Also, enforcing a non-smoking workplace can help them to emerge as industry leaders and reap business benefits with a healthier workforce and corporate image [34]. Countries like Ireland, New Zealand, Scotland, and Uruguay have built on the implementation of smoke-free laws at the local and subnational levels. With almost universal success, they have enacted and implemented laws to protect workers and the public from SHS in almost all indoor workplaces that have been applauded by the scientific and general community all over the world [35].

In our study, still, 28.9% of non-smoker males and 17.8% of non-smoker females were exposed to SHS at the workplace as per GATS-II. The high exposure among men could be due to the male-dominated workforce in India. Further, a cross-country comparison of SHS exposure done by King et al. stated that in every country, workplace SHS exposure was higher among males than females [36]. The SHS exposure at the workplace was high among those who were not employed with unorganized sectors or among self-employed. For the same reason and poor implementation of COTPA, the exposure is high in rural areas compared to urban areas [37]. In addition, the protective measures provided are partial and inadequate and do not provide a comprehensive smoke-free environment in any workplace.

Based on our analysis, though the prevalence of SHS exposure in public places has decreased, it is still affecting one out of every three non-smokers. India is currently in the process of increasing compliance with no-smoking laws in public places, and this study reinforces

the urgent need for improving compliance in India. As younger and male non-smokers are highly exposed to SHS who are prone to initiating smoking in the future, strict enforcement of legislation is needed in all around the country. Comprehensive smoke-free policies have been shown to substantially reduce SHS exposure among non-smokers working in public places and other customers visiting these public places [23]. We infer that the knowledge of the harmful effect of smoking and SHS does not necessitate behavior, similar to observations made in other studies [11].

Knowledge about the harmful effects of smoking has reduced among non-smokers exposed to SHS at home and public places from GATS-1 to GATS-II. It could be an important missed opportunity by the NTCP, whose approach is primarily regulatory instead of participatory. Though the knowledge of harmful effects of smoking has improved among participants exposed in the workplace, the observed change is minimal. Knowledge is a source of empowerment that enables the non-smokers to implement no-smoking rules/norms in their homes, and surroundings more stringently. However, we could not assess the reasons for the decrease in knowledge from GATS round I to II instead of high media coverage through advertisements and warnings. Future studies can try to assess any kind of selective media preferences regarding the place of anti-tobacco campaigning and practice the client segment approach in a more participatory model.

Despite the positive results through COTPA enforcement, concerns have been verbalized regarding the potential displacement of smoking from the workplace and public places into the smokers' homes, and this is supported by few studies [38–40]. However, some other studies have also observed an encouraging reverse trend, i.e., smoke-free legislation decrease smoking in homes, and such legislation may motivate smokers to give up smoking in their own homes [22,41]. Therefore, we also need to address the needs of smokers as most of them need assistance to quit, though they need a different approach.

There are certain limitations to the study. The cross-sectional nature of the data prevents us from analyzing temporal trends. As it is secondary data analysis, we could not assess the reasons for the decrease in the knowledge of non-smoker respondents regarding the harmful effect of tobacco. It is was not possible to quantify the impact of each initiative taken by the government between the two rounds of GATS in reducing SHS exposure. There is some difference in how the survey has been implemented in GATS I and II, including some changes in operational definitions. However, the effect of this change on the results of this analysis is negligible. Finally, we could not provide the prevalence of SHS exposure among children ≤15 years at home and public places as they were not included in the survey. Future surveys can include family-related characteristics like total family members and a number of children in the family to assess the SHS exposure among them.

## Conclusions

We observed a significant reduction in the prevalence of exposure to SHS in-home and public places, with negligible change at workplaces in India between GATS-I and GATS-II. High SHS exposure among females and youth is a matter of concern. The study calls for focused, multisectoral, and community-based interventions in India in addition to the stringent implementation of anti-tobacco legislation in protecting non-smokers from SHS exposure.

## Author Contributions

**Conceptualization:** Madhur Verma, Soundappan Kathirvel, Sonu Goel.

**Data curation:** Madhur Verma, Milan Das.

**Formal analysis:** Madhur Verma.

**Investigation:** Milan Das.

**Methodology:** Soundappan Kathirvel, Sonu Goel.

**Software:** Madhur Verma, Milan Das.

**Supervision:** Soundappan Kathirvel, Sonu Goel.

**Visualization:** Sonu Goel.

**Writing – original draft:** Madhur Verma, Soundappan Kathirvel, Milan Das.

**Writing – review & editing:** Madhur Verma, Soundappan Kathirvel, Ramnika Aggarwal, Sonu Goel.

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
