## [Decision Letter · Decision Letter 0]

11 Mar 2020

PONE-D-19-35960

Trends and pattern of second-hand smoke exposure amongst the non-smokers in India-A secondary data analysis from Global Adult Tobacco Survey (GATS) I & II

PLOS ONE

Dear Dr. Goel,

Thank you for submitting your manuscript to PLOS ONE. After careful consideration, we feel that it has merit but does not fully meet PLOS ONE’s publication criteria as it currently stands. Therefore, we invite you to submit a revised version of the manuscript that addresses the points raised during the review process.

In addition to making the changes that the reviewers requested, please include quantitative findings in the Results section of the abstract.

We would appreciate receiving your revised manuscript by Apr 25 2020 11:59PM. To enhance the reproducibility of your results, we recommend that if applicable you deposit your laboratory protocols in protocols.io, where a protocol can be assigned its own identifier (DOI) such that it can be cited independently in the future. For instructions see: http://journals.plos.org/plosone/s/submission-guidelines#loc-laboratory-protocols

We look forward to receiving your revised manuscript.

Kind regards,

Stanton A. Glantz

Academic Editor

PLOS ONE

Journal Requirements:

Reviewers' comments:

Reviewer's Responses to Questions

**Comments to the Author**

1. Is the manuscript technically sound, and do the data support the conclusions?

Reviewer #1: Yes

Reviewer #2: Yes

2. Has the statistical analysis been performed appropriately and rigorously? 

Reviewer #1: Yes

Reviewer #2: Yes

3. Have the authors made all data underlying the findings in their manuscript fully available?

Reviewer #1: Yes

Reviewer #2: Yes

4. Is the manuscript presented in an intelligible fashion and written in standard English?

Reviewer #1: Yes

Reviewer #2: Yes

5. Review Comments to the Author

Reviewer #1: The observation that respondents' knowledge of the health effects of SHS decreased from 2009 to 2016 is a matter of some concern. The possible explanations in the discussion section of "media preferences" and "sample selection bias" are not justified and do not seem plausible. What were the differences in tobacco regulation and enforcment, health promotion and public education from one period to the next?

Reviewer #2: The manuscript "Trends and pattern of second-hand smoke exposure amongst the non-smokers in India-A secondary data analysis from Global Adult Tobacco Survey (GATS) I & II" uses data from 2 repeated cross-sectional surveys to assess changes in secondhand smoke exposure, including a pooled analysis of the 2 surveys to examine factors associated with SHS exposure. This manuscript continues to add to the growing body of literature regarding SHS, and is particularly important given the data are from India. It has important policy implications. Below are some points that are meant to help improve the manuscript, and are needed prior to publication.

1. Overall, there is a technical edit. For instance, the use of the term maximum and minimum rather than highest and lowest. Also there are times when a period appears mid-sentence. Suggest having a careful read of the manuscript.

2. Introduction, line 94 why is this 48% here, but apparently 57% in line 92?

3. Introduction, line 96 suggest using term in contrast rather than constrastingly

4. More details are needed in the statistical analyses; why was a pooled analysis done? Overall the methods, could be more succinct. For instance you could have the operational definition with variables on SHS. The sample selection could be reduced to 1-2 lines in the statistical analysis, with information in a table on the sample size.

5. Line 198, why are these knowledge variables in sociodemographic factor list?

6. Consistency is needed in the use of commas for numbers in the thousands. Further, consistency is needed regarding the numbers after a decimal; some results have decimals to the tenth spot, others to the hundredths or thousandths, or no decimal spots.

7. Results, line 204 Add percentages to the 11262 & 12475.

8. Results, line 206 What is this a percentage of? Above you say 57,813 is 83.42% but this is only decreased by 113, how did it drop to 47.2%

9. Table 1, for weighted prevalence estimates, suggest having 95% CI

10. Table 1, see comment 6 about the decimal spots

11. Table 1, the chi-square value seem to indicate difference within survey year, but there is no indication if there are differences between years?

12. Table 1, shouldn't the denominator for all the places be 11,262 and 12,475? How can the denominator be the same as the public places? I would think that workplace is the limiting factor?

13. Table 1, these knowledge variables are concerning and not discussed at all until the discussion. In fact if seems that people have less knowledge about consequences of smoking over time? This seems important, and should be discussed more?

14. You have a pooled analysis as well as separate analyses for the two surveys. This isn't clear in the methods.

15. The discussion is well written, and compares results to current literature.

16. There are a few times that the authors mention a pooled analysis for policy implications, but it is not clear why that is the case.

6. PLOS authors have the option to publish the peer review history of their article (what does this mean?). If published, this will include your full peer review and any attached files.

Reviewer #1: No

Reviewer #2: No

---

## [Author Response · Author response to Decision Letter 0]

23 Apr 2020

Response to the comments

Query 1. Reviewer #1: The observation that respondents' knowledge of the health effects of SHS decreased from 2009 to 2016 is a matter of some concern. The possible explanations in the discussion section of "media preferences" and "sample selection bias" are not justified and do not seem plausible. What were the differences in tobacco regulation and enforcement, health promotion and public education from one period to the next?

Response: Thank you for your comment to improve our manuscript. 

We have now written it more explicitly that “It could be an important missed opportunity by the NTCP, whose approach is primarily regulatory instead of participatory. Though the knowledge of harmful effects of smoking has improved among participants exposed in the workplace, the observed change is minimal. Knowledge is a source of empowerment that enables the non-smokers to implement no-smoking rules/norms in their homes, and surroundings more stringently.” 

Also we have mentioned that “However, we could not assess the reasons for the decrease in knowledge from GATS round I to II instead of high media coverage through advertisements and warnings. Future studies can try to assess any kind of selective media preferences regarding the place of anti-tobacco campaigning and practice the client segment approach in a more participatory model.”

We have also added text pertaining to the changes observed from GATS I to GATS II in the discussion section. 

Page 19-22

Line 277-291, 348-358

Reviewer #2:

Query 1. The manuscript "Trends and pattern of second-hand smoke exposure amongst the non-smokers in India-A secondary data analysis from Global Adult Tobacco Survey (GATS) I & II" uses data from 2 repeated cross-sectional surveys to assess changes in second-hand smoke exposure, including a pooled analysis of the 2 surveys to examine factors associated with SHS exposure. This manuscript continues to add to the growing body of literature regarding SHS, and is particularly important given the data are from India. It has important policy implications. Below are some points that are meant to help improve the manuscript, and are needed prior to publication.

Response: We agree with the reviewer’s observations and thank you for your constructive comments. This manuscript has now been aptly revised in view of the comments by expert reviewers and we are sure that it will continue to add to the growing body of literature regarding SHS.

Query 2. Overall, there is a technical edit. For instance, the use of the term maximum and minimum rather than highest and lowest. Also, there are times when a period appears mid-sentence. Suggest having a careful read of the manuscript. 

Response: We have now extensively revised our manuscript, and we are hopeful that it doesn’t disappoint you anymore.

Query 3. Introduction, line 94 why is this 48% here, but apparently 57% in line 92?

Response: we have now revised data and removed any discrepancy throughout the manuscript. 

Page no. 5

Line : 94-98

Query 4. Introduction, line 96 suggest using term in contrast rather than constrastingly

Response: we have modified it as per your suggestions. 

Page no. 5

Line: 98-99

Query 5. More details are needed in the statistical analyses; why was a pooled analysis done? Overall the methods, could be more succinct. For instance you could have the operational definition with variables on SHS. The sample selection could be reduced to 1-2 lines in the statistical analysis, with information in a table on the sample size.

Response: We have revised the methodology part and concise it as per your comments. We have now added the operational definitions and sample selection process under the common heading: “Operational definitions and sample selection.” 

Also, we have now added Fig.1& 2 to depict sample selection procedure for our study from GATSI & GATS II datasets and observed prevalence of SHS exposure. 

Regarding the pooled analysis: We initially did pooled analysis so that we can have a larger sample size of people who were exposed to SHS at all three places- and then extended it to individual categories (table 2-3-4) to maintain uniformity of tables. But as all the reviewers had raised their comments on this pooled analysis- we have now decided to remove this part. Also, we also feel that it dilutes the efforts of highlighting the trends that we intend to assess from GATS I to GATS II. 

We thank the reviewers for this suggestion.

Page no. 7

Line: 129-155, and Figure1 &2 as separate attachment files.

Query 6. Line 198, why are these knowledge variables in sociodemographic factor list?

Response: We have now rephrased the paragraph

Page no.8-9

Line: 175-180

Query 7. Consistency is needed in the use of commas for numbers in the thousands. Further, consistency is needed regarding the numbers after a decimal; some results have decimals to the tenth spot, others to the hundredths or thousandths, or no decimal spots.

Response: we have now revised our manuscript extensively in compliance to other comments. 

Query 8. Results, line 204 Add percentages to the 11262 & 12475.

Response: we have revisited our datasets and redone our analysis. We have added % wherever necessary

Page no. 9

Line: 187-196

Query 9. Results, line 206 What is this a percentage of? Above you say 57,813 is 83.42% but this is only decreased by 113, how did it drop to 47.2%

Response: we have revisited our datasets and redone our analysis. We have added % wherever necessary

Page no. 9

Line: 187-196

Query 10. Table 1, for weighted prevalence estimates, suggest having 95% CI

Response: we have now added 95% CI

Page no. 11

Table 1

Query 11. Table 1, see comment 6 about the decimal spots

Response: we have revised our tables now and made them look more uniform throughout. 

Page no. 11, 14-17

Tables 1-5. 

Query 12. Table 1, the chi-square value seem to indicate difference within survey year, but there is no indication if there are differences between years?

Response: we have added chi-square to indicate differences between the years. Also, to keep it simple we have removed chi-square that indicated differences within the surveys to look for differences in the surveys was not our objective. 

Page no. 11

Table 1

Query 13. Table 1, shouldn't the denominator for all the places be 11,262 and 12,475? How can the denominator be the same as the public places? I would think that workplace is the limiting factor?

Response: we have revisited our analysis. We have modified the figures. We sincerely thank you for raising this query

Page no. 11

Table 1

Query 14. Table 1, these knowledge variables are concerning and not discussed at all until the discussion. In fact if seems that people have less knowledge about consequences of smoking over time? This seems important, and should be discussed more?

Response: we have now discussed changes in the knowledge levels of non-smokers more explicitly in our results and discussion section. 

Page no. 10, 21, 22

Line: 205-206, 212-213, 218 -220 and 348-358

Query 15. You have a pooled analysis as well as separate analyses for the two surveys. This isn't clear in the methods.

Response: We initially did pooled analysis so that we can have a larger sample size of people who were exposed to SHS at all three places- and then extended it to individual categories (table 2-3-4) to maintain uniformity of tables. But as all the reviewers had raised their comments on this pooled analysis- we have now decided to remove this part. Also, we also feel that it dilutes the efforts of highlighting the trends that we intend to assess from GATS I to GATS II. So we now removed all the mentions of Pooled analysis from our manuscript. 

We thank the reviewers for this suggestion.

Query 16. The discussion is well written, and compares results to current literature.

Response: we sincerely thank you for your encouraging words 

Query 17. There are a few times that the authors mention a pooled analysis for policy implications, but it is not clear why that is the case.

Response: We have now removed the pooled analysis part. However, we have precisely added the policy changes observed from GATS I to GATS II in the study while discussing the SHS exposure at the home, workplace and public places. 

Page no. 19-20

Line: 277-291

---

## [Decision Letter · Decision Letter 1]

14 May 2020

Trends and patterns of second-hand smoke exposure amongst the non-smokers in India-A secondary data analysis from the Global Adult Tobacco Survey (GATS) I & II

PONE-D-19-35960R1

Dear Dr. Goel,

We are pleased to inform you that your manuscript has been judged scientifically suitable for publication and will be formally accepted for publication once it complies with all outstanding technical requirements.

With kind regards,

Stanton A. Glantz

Academic Editor

PLOS ONE

Additional Editor Comments (optional):

Reviewers' comments:

Reviewer's Responses to Questions

**Comments to the Author**

1. If the authors have adequately addressed your comments raised in a previous round of review and you feel that this manuscript is now acceptable for publication, you may indicate that here to bypass the “Comments to the Author” section, enter your conflict of interest statement in the “Confidential to Editor” section, and submit your "Accept" recommendation.

Reviewer #2: All comments have been addressed

2. Is the manuscript technically sound, and do the data support the conclusions?

Reviewer #2: Yes

3. Has the statistical analysis been performed appropriately and rigorously? 

Reviewer #2: Yes

4. Have the authors made all data underlying the findings in their manuscript fully available?

Reviewer #2: Yes

5. Is the manuscript presented in an intelligible fashion and written in standard English?

Reviewer #2: Yes

6. Review Comments to the Author

Reviewer #2: (No Response)

7. PLOS authors have the option to publish the peer review history of their article (what does this mean?). If published, this will include your full peer review and any attached files.

Reviewer #2: No

---

## [Editor Report · Acceptance letter]

28 May 2020

PONE-D-19-35960R1 

Trends and patterns of second-hand smoke exposure amongst the non-smokers in India-A secondary data analysis from the Global Adult Tobacco Survey (GATS) I & II 

Dear Dr. Goel:

I am pleased to inform you that your manuscript has been deemed suitable for publication in PLOS ONE. Congratulations! Your manuscript is now with our production department. 

With kind regards,

on behalf of

Professor Stanton A. Glantz 

Academic Editor

PLOS ONE